# Dataset Documenting the Interactions of Biochar with Manure, Soil, and Plants: Towards Improved Sustainability of Animal and Crop Agriculture

Darcy Bonds [1], Jacek A. Koziel [2,*], Mriganka De [3,4], Baitong Chen [2], Asheesh K. Singh [3] and Mark A. Licht [3]

1 Department of Horticulture, Iowa State University, Ames, IA 50011, USA; dgbonds@iastate.edu
2 Department of Agriculture and Biosystems Engineering, Iowa State University, Ames, IA 50011, USA; baitongc@iastate.edu
3 Department of Agronomy, Iowa State University, Ames, IA 50011, USA; mriganka.de@mnsu.edu (M.D.); singhak@iastate.edu (A.K.S.); lichtma@iastate.edu (M.A.L.)
4 Department of Biological Sciences, Minnesota State University, Mankato, MN 50011, USA
* Correspondence: koziel@iastate.edu

**Abstract:** AbstractPlant and animal agriculture is a part of a larger system where the environment, soil, water, and nutrient management interact. Biochar (a pyrolyzed biomass) has been shown to affect the single components of this complex system positively. Biochar is a soil amendment, which has been documented for its benefits as a soil enhancer particularly to increase soil carbon, improve soil fertility, and better nutrient retention. These effects have been documented in the literature. Still, there is a need for a broader examination of these single components and effects that aims at the complementarity and synergy attainable with biochar and the animal and crop-production system. Thus, we report a comprehensive dataset documenting the interactions of biochar with manure, soil, and plants. We evaluated three biochars mixed with manure alongside both manure and soil controls for improvement in soil quality, reduction in nutrient movement, and increase in plant nutrient availability. We explain the experiments and the dataset that contains the physicochemical properties of each biochar–manure mixture, the physicochemical properties of soil amended with each biochar–manure mixture, and the biomass and nutrient information of plants grown in biochar–manure mixture-amended soil. This dataset is useful for continued research examining both the short- and long-term effects of biochar–manure mixtures on both plant and soil systems. In addition, these data will be beneficial to extend the findings to field settings for practical and realized gains.

**Dataset:** Submitted to be published as a supplement to this paper in the journal Data.

**Dataset License:** CC-BY.

**Keywords:** carbon cycling; nutrient cycling; soil amendment; manure; biochar; corn; maize; soybeans; fertilizer

## 1. Summary

Plant and animal agriculture is a part of a larger system where the environment, soil, water, and nutrient management interact. Biochar (a pyrolyzed biomass) has been shown to affect the single components of this complex system positively. These effects have been documented in the literature. Over 40 published reviews and papers document interactions between biochar and soil, focusing primarily on decontamination [1], amendment of soil properties [2], and carbon sequestration [3]. Only half of these published papers and reviews discuss the interactions between biochar and manure, highlighting organic waste composting systems [4], greenhouse gas emissions [5], and manure decontamination [6]. About fifteen reviews and papers explore the relationship between biochar and plants,

mainly discussing detoxification [7], nutrient dynamics [8], and crop productivity [9]. Still, there is a need for a broader examination of these single components and effects that aims at the complementarity and synergy attainable with biochar and the animal and crop-production system.

Recently, we provided a more comprehensive story concerning the interactions between a system in which biochar, soil, manure, and plants are involved and interacting [10,11]. We explored the possibility of a holistic approach, i.e., using a biochar–manure mixture as fertilizer in soil-plant systems. Specifically, we focused on improving nutrient recycling, solving livestock odor problems, and increasing crop yields (Figure 1) [10,11]. This proposed concept can be classified as a Technology Readiness Level 2 or Level 3 [12], as the concept of this study has been validated and proven through its supported experiments [10,11]. Scaling up, demonstration, and proven effectiveness in greater number of field-case studies is still needed.

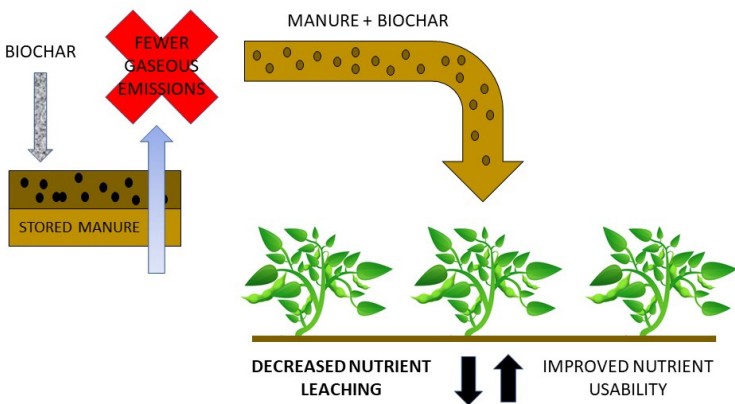

**Figure 1.** Concept of sustainable animal and crop-production system (redrawn after Figure 1 in [11] under CC BY License).

However, the experimental design for [10,11] contained a wider scope of environmental and agronomic data that were not included in these published works apart from some discussion of potentially beneficial trends. Thus, we are publishing this dataset so that the data not included in the published work will become available for consideration concerning these emerging trends, which could improve sustainable practices in environmental, crop agriculture, and animal systems.

The data collected from this investigation of the interactions between the soil, plant, biochar, and manure elements of an agricultural system revealed a positive impact. Specifically, soil composition of organic matter, C, N, P, and K significantly increased [10]. Additionally, an increase of organic matter and C in the soil was observed in the greenhouse setting without affecting plant nutrient uptake [11]. While only swine manure was used [10,11], it would be interesting to consider manure from different types of animal production and manure-management systems and the effects on a biochar–manure mixture. In the future, studies investigating the interactions between these factors in a field setting over a longer period of time, with the inclusion of the manure type variable, will reveal more about these animal and crop-production systems.

## 2. Data Description

This dataset is organized as a spreadsheet with a total of six sheets (Table 1). The first sheet defines abbreviations and terms used throughout the dataset as well as a brief summary of each successive sheet. The second sheet contains the physicochemical properties of the control treatments and manure–biochar mixtures before soil application. The third sheet contains the soil physicochemical of the control and experimental treatments. The fourth sheet contains the data and calculations necessary to determine the nutrient efficiency ratio for each nutrient analyzed in corn (*Zea mays* L.)/soybean (*Glycine max* L.) biomass. The

fifth sheet contains initial biomass data taken before biomass physicochemical analysis. The sixth sheet contains the amount of each macro/micronutrients found in corn/soybean biomass. In this dataset, the character 'S' denotes a sample of soil containing a soybean plant, and the character 'C' denotes a sample of soil containing a corn plant.

**Table 1.** Summary of content organized in (Supplementary Material 'Biochar_Manure_Soil_Plant_Interactions.xlsx') data set for Midwest soil, treatments, and plants.

| Sheet Name | Content Description |
|---|---|
| READ_ME | Treatment abbreviation guide <br> Sheet navigation guide |
| Initial Treatment Data | Moisture content <br> Organic matter content <br> Nitrate and ammonium content <br> Mineral/nutrient content <br> pH <br> Carbon/Nitrogen ratio |
| Soil Data | Total exchange capacity <br> pH <br> Organic matter content <br> Anion content <br> Cation content <br> Cation base saturation <br> Extractable minors <br> Nitrate and ammonium content <br> Carbon/Nitrogen profile |
| Nutrient Efficiency Ratio | Units of yield <br> Units of elements in tissue <br> Nutrient efficiency ratio |
| Biomass Data | Biomass weight <br> Plant growth stage <br> Nitrogen content <br> Phosphorus content |
| Plant Data | Nutrient content of plant biomass |

### 2.1. READ_ME

This sheet lists the treatment symbols and abbreviations used throughout the dataset spreadsheet (Treatment Abbreviation Guide) as well as a brief outline of the successive sheets along with method and equation citations (Sheet Guide). Table 2 provides a guide of what content can be found in the 'READ_ME' sheet.

**Table 2.** Summary and guide of content in the 'READ_ME' sheet.

| Content | Location |
|---|---|
| Experimental treatment abbreviation key | Top left textbox |
| Greenhouse treatment abbreviation key | Top right textbox |
| Sheet navigation guide | Bottom textbox |

### 2.2. Initial Treatment Data

This sheet contains the physicochemical properties of manure control and tested manure–biochar mixtures along with calculated averages and standard deviations among replicates before their use in the experiment. The properties of pure swine manure not mixed with biochar (M) are located in Columns B and C, with averages and standard deviations in columns D and E. Due to a lab error, the manure control group only contains two replicates as opposed to the three replicates of the other experimental treatments. The properties of swine manure mixed with red oak biochar (MRO) are located in Columns F

through H, with averages and standard deviations in Columns I and J. The properties of swine manure mixed with highly alkaline porous corn stover biochar (MHAP) are located in Columns K through M, with averages and standard deviations located in Columns N and O. The properties of swine manure mixed with highly alkaline porous corn stover biochar (MHAPE) engineered with iron are located in Columns P through R, with averages and standard deviations located in Columns S and T. Table 3 provides an example of what content can be found in the 'Initial Treatment Data' sheet.

**Table 3.** An example of the content that can be found in the 'Initial Treatment Data' sheet. More detailed information for all replicates is included in (Supplementary Material 'Biochar_Manure_Soil_Plant_Interactions.xlsx').

| Physicochemical Characteristic | M | MRO | MHAP | MHAPE |
|---|---|---|---|---|
| Moisture (% wet) | - | 21.1 | 57.5 | 50.9 |
| LOI Organic Matter (% wet) | - | 43.4 | 30.2 | 26.5 |
| $NH_4$-N (% wet) | 0.1 | 0.01 | 0.02 | 0.5 |
| $NO_3$-N (% wet) | 0 | 0 | 0 | 0 |
| Mn (ppm wet) | 35.4 | 62.4 | 60.9 | 261.0 |
| pH | - | 9.9 | 9.7 | 7.6 |
| Carbon–Nitrogen Ratio | - | 46.6 | 26.4 | 12.7 |

The authors would like to stress the importance of standardization and its place in the use of this data for agriculture applications. Initiatives, such as the International Biochar Initiative (IBI) [13] and the European Biochar Certificate [14], are examples of available frameworks for biochar standardization aiming for established guidelines for feedstock, process, basic elemental content, and thresholds for targeted toxics.

*2.3. Soil Data*

This sheet contains the physicochemical properties along with calculated averages and standard deviations of baseline soil (Row 3), soil from soil columns (Rows 5 through 33), as well as greenhouse pots containing corn (Rows 35 through 68) and soybean (Rows 70 through 103) plants. Properties of untreated soil (SOIL) are located in Rows 11 through 15, Rows 35 through 40, and Rows 70 through 75. Properties of soil treated with swine manure and no biochar (M) are located in Rows 5 through 9, Rows 42 through 47, and Rows 77 through 82. Properties of soil treated with manure and red oak biochar (MRO) are located in Rows 17 through 21, Rows 49 through 54, and Rows 84 through 89. Properties of soil treated with manure and highly alkaline porous corn stover biochar (MHAP) are located in Rows 23 through 27, Rows 56 through 61, and Rows 91 through 96. Properties of soil treated with manure and highly alkaline porous corn stover biochar engineered with iron (MHAPE) are located in Rows 29 to 33, Rows 63 through 68, and Rows 98 through 103. Table 4 provides an example of what content can be found in the 'Soil Data' sheet.

**Table 4.** An example of the content that can be found in the 'Soil Data' sheet. More detailed information for all replicates is included in (Supplementary Material 'Biochar_Manure_Soil_Plant_Interactions.xlsx').

| Sample ID | Total Exchange Capacity (ME/100 g) | pH | Organic Matter (%) | Mehlich III (ppm) | Ca (ppm) | Ca (%, Base Saturation) | Fe (ppm) | $NO_3$-N (ppm) | Carbon–Nitrogen Ratio |
|---|---|---|---|---|---|---|---|---|---|
| Baseline | 18.0 | 7.6 | 2.8 | 29 | 2296 | 63.9 | 137 | 7.1 | 11.1 |
| M | 16.6 | 7.5 | 3.2 | 58 | 1994 | 60.2 | 484 | 33.7 | 13.4 |
| SOIL | 17.1 | 7.5 | 3.0 | 36 | 2149 | 62.8 | 506 | 15.1 | 10.8 |
| MRO | 16.7 | 7.5 | 3.3 | 51 | 2022 | 60.4 | 459 | 18.8 | 11.3 |
| MHAP | 15.8 | 7.5 | 3.4 | 43 | 1919 | 60.8 | 447 | 22.2 | 11.9 |
| MHAPE | 16.9 | 7.3 | 3.4 | 39 | 2038 | 60.4 | 475 | 36.8 | 12.6 |

## 2.4. Nutrient Efficiency Ratio

Nutrient efficiency ratio is found by dividing the units of yield (found in Column C), or total biomass, by the units of the element of interest found in the harvested plant tissue (found in Columns F through L). Nutrient efficiency ratios were calculated for the three manure–biochar treatments along with the two experimental controls for both corn (found in Rows 5 through 24) and soybean (found in Rows 29 through 48) trials. Individual nutrient efficiency ratios along with calculated averages and standard deviations were recorded for nitrogen (Columns N through P), phosphorus (Columns Q through S), potassium (Columns T through V), calcium (Columns W through Y), magnesium (Columns Z through AB), sulfur (Columns AC through AE), and carbon (Columns AF through AH). Table 5 provides an example of what content can be found in the 'Nutrient Efficiency Ratio' sheet.

**Table 5.** An example of the content that can be found in the 'Nutrient Efficiency Ratio' sheet. More detailed information for all replicates is included in (Supplementary Material 'Biochar_Manure_Soil_ Plant_Interactions.xlsx').

| Pot ID | Biomass (g) | N (g in Tissue) | N (NER) |
|--------|-------------|-----------------|---------|
| C1 | 3.0 | 0.06 | 47.8 |
| C2 | 4.2 | 0.06 | 68.0 |
| C3 | 3.4 | 0.05 | 64.9 |
| C4 | 2.8 | 0.04 | 78.1 |
| S1 | 1.7 | 0.06 | 29.1 |
| S2 | 0.9 | 0.05 | 19.7 |
| S3 | 1.0 | 0.04 | 27.0 |
| S4 | 0.8 | 0.04 | 21.6 |

## 2.5. Biomass Data

This sheet contains initial biomass data taken before corn (Columns A through K) and soybean (Columns M through R) plant nutrient analysis, including the mass of bag + biomass (Columns C and O) and biomass alone with calculated averages and standard deviations for experimental treatments and controls (Columns D through F and Columns P through R). Additional data were collected for corn plants, specifically corn plant growth stage at the time of nutrient analysis, along with calculated averages for each pot and each experimental treatment (Columns G through J) and the standard deviation of experimental treatment averages (Column K). Table 6 provides an example of what content can be found in the 'Biomass Data' sheet.

**Table 6.** An example of data that can be found in the 'Biomass Data' sheet. More detailed information for all replicates is included in (Supplementary Material 'Biochar_Manure_Soil_Plant_ Interactions.xlsx').

| Pot ID | Biomass (g) | Plant Growth Stage |
|--------|-------------|--------------------|
| C1 | 3.0 | V6 |
| C2 | 4.2 | V6 |
| C3 | 3.4 | V6 |
| C4 | 2.8 | V6 |
| S1 | 1.7 | - |
| S2 | 0.9 | - |
| S3 | 1.0 | - |
| S4 | 0.8 | - |

## 2.6. Plant Data

This sheet contains the elemental nutrient content of analyzed biomass for all pots and all experimental treatments. Data for plants grown in untreated soil are located in Rows 4 through 9. Data for plants grown in soil treated with swine manure only are located in Rows 10 through 15. Data for plants grown in soil treated with swine manure and red oak

biochar are located in Rows 16 through 21. Data for plants grown in soil treated with swine manure and highly alkaline porous corn stover biochar are located in Rows 22 through 27. Data for plants grown in soil treated with swine manure and highly alkaline porous biochar engineered with iron are located in Rows 28 through 33. Amounts of nitrogen, phosphorus, potassium, calcium, magnesium, sulfur, and carbon were recorded as percentages for both corn (Columns B through H) and soybean (Columns R through X), along with calculated treatment averages and standard deviations. Amounts of boron, iron, manganese, copper, zinc, aluminum, and sodium were recorded as parts per million for both corn (Columns I through O) and soybean (Columns Y through AE), along with treatment averages and standard deviations. Table 7 provides an example of what content can be found in the 'Plant Data' sheet.

**Table 7.** An example of data that can be found in the 'Plant Data' sheet. More detailed information for all replicates is included in (Supplementary Material 'Biochar_Manure_Soil_Plant_Interactions.xlsx').

| Pot ID | N (%) | P (%) | K (%) | B (ppm) | Fe (ppm) | Mn (ppm) |
|--------|-------|-------|-------|---------|----------|----------|
| C1 | 2.09 | 0.14 | 2.56 | 11.90 | 50.20 | 37.70 |
| C2 | 1.47 | 0.13 | 2.34 | 12.40 | 47.20 | 40.80 |
| C3 | 1.54 | 0.15 | 2.41 | 14.30 | 58.50 | 50.30 |
| C4 | 1.28 | 0.15 | 2.41 | 15.10 | 38.90 | 41.70 |
| S1 | 3.44 | 0.20 | 2.31 | 44.50 | 69.90 | 93.10 |
| S2 | 5.08 | 0.30 | 2.45 | 39.70 | 77.90 | 102.00 |
| S3 | 3.70 | 0.25 | 2.51 | 38.90 | 62.50 | 89.10 |
| S4 | 4.62 | 0.32 | 2.49 | 39.30 | 72.80 | 107.00 |

## 3. Methods

Properties of the biochars used in this experiment were described in detail in [10]. Briefly, three biochars were used: corn stover autothermal alkaline porous (HAP, pH = 9.2), corn stover autothermal porous Fe-engineered (HAPE, pH = 5.4), and red oak (RO, pH = 7.5). Conventional fast pyrolysis (for RO) and autothermal pyrolysis (for HAP and HAPE) was used as described in detail by Polin et al., 2019 [15] and Rollag et al., 2020 [16]. The biochar properties, such as C (fixed and total), total N, ash content, moisture, and volatile matter, were determined with methods described elsewhere [17]. Swine manure collected from deep-pit storage was incubated for four weeks before the creation of the biochar–manure mixtures. Following assembly of each mixture, there was another four-week incubation period [10,11]. A well-drained Hanlon soil was used. Hanlon soils are of coarse-loamy texture and the mesic Cumulic Hapludoll order [10,11].

The first part of this experiment concerned a soil column study investigating interactions between biochar, manure, and soil. A more detailed account of the experimental methods for this investigation can be found in [10]. Briefly, the mixture of biochar and manure moisture, total C, total N, organic matter, $NO_3$-N, $NH_4$-N, and plant-available P/K/other nutrients were analyzed by standard methods [18–26].

As shown in Figure 2, a soil column was filled for each experimental replicate containing a soil control, manure control, or biochar–manure mixture evaluated in the experiment. Soil columns were leached periodically over a period of thirty days. Leachate was collected in sample jars and stored and analyzed for physicochemical properties along with the soil.

The second part of this experiment involved a greenhouse study exploring the interactions between biochar, manure, soil, and corn and soybean plants. A more detailed account of the experimental methods for this investigation can be found in [11]. Briefly, the plant growth stage for corn and soybean maturity was determined by standard methods [27,28].

Four replicates of each control and biochar–manure mixture investigated were used to grow both corn and soybean plants. Plants were watered with deionized water every other day over a period of nine weeks. Plant biomass was then harvested, dried, and analyzed for nutrient content, nutrient-use efficiency, and other physicochemical properties. Soil was also analyzed for physicochemical properties with standard methods [18–26]. Figure 3 shows the concept diagram illustrating the design of the experiment [11].

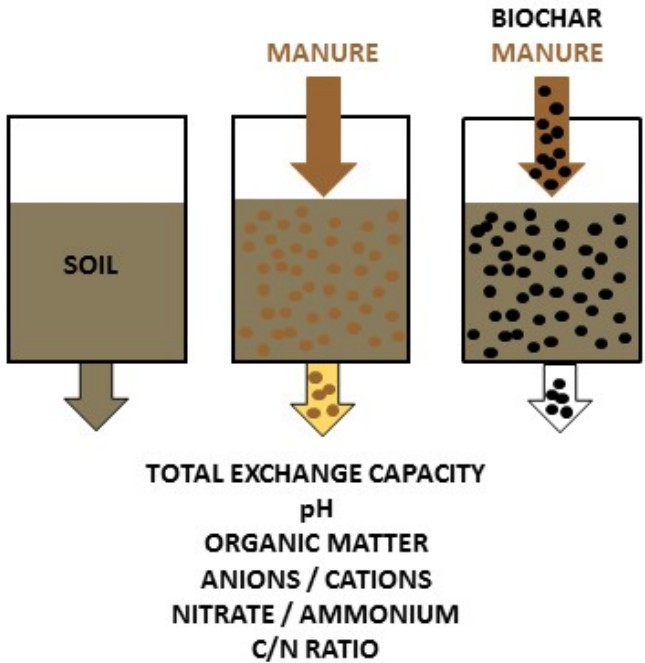

**Figure 2.** Concept diagram illustrating the design of the experiment published in [10]. Baseline soil, soil with manure added, and soil with a biochar–manure mixture added were leached and analyzed for various physicochemical properties.

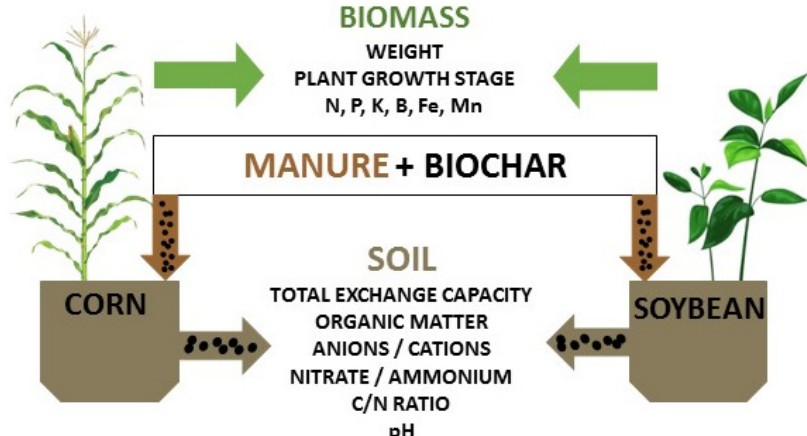

**Figure 3.** Concept diagram illustrating the design of the experiment [11]. Corn and soybeans were grown in soil (baseline; control), soil with manure added, and soil with a biochar–manure mixture added. Physicochemical properties of both plant biomass and soil were analyzed.

## 4. User Notes

The attached spreadsheet (Biochar_Manure_Soil_Plant_Interactions.xlsx) contains a well-organized data set. The first sheet (READ_ME) contains detailed instructions on how to use and navigate the data set.

**Supplementary Materials:** The following supporting information can be downloaded at: https://www.mdpi.com/article/10.3390/data7030032/s1, i.e., 'Biochar_Manure_Soil_Plant_Interactions.xlsx'.

**Author Contributions:** Conceptualization, D.B. and J.A.K.; software, D.B.; validation, D.B. and J.A.K.; resources, D.B. and J.A.K.; data curation, D.B. and J.A.K.; writing—original draft preparation, D.B.; writing—review and editing, D.B., J.A.K., B.C., A.K.S., M.D. and M.A.L.; visualization, D.B. and J.A.K.; supervision, J.A.K.; project administration, J.A.K.; funding acquisition, J.A.K., A.K.S. and M.A.L. All authors have read and agreed to the published version of the manuscript.



**Funding:** The authors are thankful to the Leopold Center for Sustainable Agriculture for 'Improving sustainability of Iowa agriculture: synergy between improved nutrient recycling, solving livestock odor problems, and crop production' grant #LCSA-AES-Koziel-2020-2. Partial funding came from Iowa State University's Freshmen Honors Program (Darcy Bonds) and the Iowa Agriculture and Home Economics Experiment Station: project number IOW05556 (Future Challenges in Animal Production Systems: Seeking Solutions through Focused Facilitation, sponsored by Hatch Act and State of Iowa funds; Jacek Koziel).

**Institutional Review Board Statement:** Not applicable.

**Informed Consent Statement:** Not applicable.

**Data Availability Statement:** Data are contained within the article and Supplementary Material.

**Acknowledgments:** The authors of this data descriptor would like to acknowledge Chumki Banik (ABE, ISU) for her leadership throughout the two published experiments from which these data were obtained. We are thankful to David Laird (Department of Agronomy, ISU) for initial consultations about the experiment and Qinglong Tian (Department of Statistics, ISU) for his help with the statistical model. The authors also thank Wyatt Murphy from the Department of Agricultural and Biosystems Engineering, Iowa State University (ABE, ISU) for building the column leachate setup, Peiyang Li and Samuel C. O'Brien (ABE, ISU) for their help with soil sampling, and Zhanibek Meirrkhanuly (ABE, ISU) for the outsourcing of biochar.

**Conflicts of Interest:** The authors declare no conflict of interest. The funders had no role in the design of the study, in the collection, analyses, or interpretation of data, in the writing of the manuscript, or in the decision to publish the results.

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
