# Peer review of "Dataset Documenting the Interactions of Biochar with Manure, Soil, and Plants: Towards Improved Sustainability of Animal and Crop Agriculture"

_data, 2022_

Round 1

Reviewer 1 Report

Review Report (Data)

Abstract

Line 26                        Replace ‘extent‘ with “extend

Summary

Line 33                     delete “.”the comma and insert                                           “and” between water and nutrient.

Line 40                        kindly delete “Only” and begin the                                      sentence with “About”

Line 43                        change aims to “aim

Lines 48 – 49              was the focus on the biochar-                                             manure mixture related to                                                   “improving nutrient cycling,                                                 solving livestock odor problems                                         and increasing crop yields” or

                                    just on crop production?

Reviewer 2 Report

The key factor, biochar quality specs is missing, most importantly PAH19,  VOC contents and physical/surface characteristics.

Experimental conditions missing, although it is suggested that this is pot trials at low TRLs. Different types of manure might get different results.

Remark: we are all aware of that there is a big difference between low TRL in lab or high TRL real life condition testing.

Reviewer 3 Report

Initially I missed the description of how the biochars were obtained, describing this procedure in the manuscript is important for the application of the evaluation in other types of soil and with other plant species, wheat, beans and cotton, for example.

Another point is what is the classification of this soil used?
I believe it is also important to mention its sand, silt and clay contents, where it was collected.
The dung is from which animal? Would the dung of another animal have a different effect? Citing the origin of the manure is also important because of the global reach that the article can reach!

I particularly felt the lack of a conclusion or final considerations regarding the manuscript. Describing future steps on the subject, what the data showed to the work team, showing ways that can further enrich the information we have on biochar are important.
